# Daily Vegetables Intake and Response to COPD Rehabilitation. The Role of Oxidative Stress, Inflammation and DNA Damage

**DOI:** 10.3390/nu13082787

**Published:** 2021-08-14

**Authors:** Sara Ilari, Laura Vitiello, Patrizia Russo, Stefania Proietti, Mirta Milić, Carolina Muscoli, Vittorio Cardaci, Carlo Tomino, Gaia Bonassi, Stefano Bonassi

**Affiliations:** 1Department of Health Science, Institute of Research for Food Safety & Health (IRC-FSH), University “Magna Graecia” of Catanzaro, 88201 Catanzaro, Italy; sara.ilari@hotmail.it (S.I.); muscoli@unicz.it (C.M.); 2Laboratory of Flow Cytometry, IRCCS San Raffaele Roma, 00166 Rome, Italy; laura.vitiello@sanraffaele.it; 3Clinical and Molecular Epidemiology, IRCCS San Raffaele Roma, 00166 Rome, Italy; Stefania.proietti@sanraffaele.it (S.P.); stefano.bonassi@sanraffaele.it (S.B.); 4Department of Human Sciences and Quality of Life Promotion, San Raffaele University, 00166 Rome, Italy; 5Mutagenesis Unit, Institute for Medical Research and Occupational Health, 10000 Zagreb, Croatia; mirtamil@gmail.com; 6Pulmonary Rehabilitation Unit, IRCCS San Raffaele Roma, 00166 Rome, Italy; vittorio.cardaci@sanraffaele.it; 7Scientific Direction, IRCCS San Raffaele Roma, 00166 Rome, Italy; carlo.tomino@sanraffaele.it; 8S.C. Medicina Fisica e Riabilitazione Ospedaliera, ASL4, Azienda Sanitaria Locale Chiavarese, 16034 Chiavari, Italy; gaia.bonni@gmail.com

**Keywords:** chronic obstructive pulmonary disease (COPD), pulmonary rehabilitation, vegetables, DNA damage, genomic instability, oxidative stress, inflammation

## Abstract

Chronic obstructive pulmonary disease (COPD) is a respiratory disease associated with airways inflammation and lung parenchyma fibrosis. The primary goals of COPD treatment are to reduce symptoms and risk of exacerbations, therefore pulmonary rehabilitation is considered the key component of managing COPD patients. Oxidative airway damage, inflammation and reduction of endogenous antioxidant enzymes are known to play a crucial role in the pathogenesis of COPD. Recently, also natural antioxidants have been considered as they play an important role in metabolism, DNA repair and fighting the effects of oxidative stress. In this paper we evaluated the response of 105 elderly COPD patients to pulmonary rehabilitation (PR), based on high or low vegetable consumption, by analyzing clinical parameters and biological measurements at baseline and after completion of the three weeks PR. We found that daily vegetable intake in normal diet, without any specific intervention, can increase the probability to successfully respond to rehabilitation (65.4% of responders ate vegetables daily vs. 40.0% of non-responders, *p* = 0.033). The association was especially evident in subjects ≥ 80 year of age (OR = 17.0; *p* < 0.019). Three weeks of pulmonary rehabilitation are probably too short to reveal a reduction of the oxidative stress and DNA damage, but are enough to show an improvement in the patient’s inflammatory state.

## 1. Introduction

Chronic obstructive pulmonary disease (COPD) is characterized by airflow limitation associated with a chronic inflammatory response to noxious stimuli or gases [1,2] and is among the most common causes of mortality and disability in the world [2,3]. COPD is age-dependent and mortality is more common in women than in men [3]. Smoking is the primary risk factor for COPD, although, only 15% of smokers develop the disease [3]; infections and obesity also are considered triggering conditions [1,2,4]. Oxidative stress and associated inflammation play a key role in the pathogenesis of obstructive airway diseases. Specifically, inhaled particles and inflammatory mediators contribute to the development of reactive oxygen and nitrogen species (free radicals, reactive oxygen species -ROS-; reactive nitrogen species-RNS), resulting in oxidative stress, responsible for harmful effects, such as damage to lipids, proteins, and nucleic acids [1,2].

It has been observed that COPD patients had an increase in markers of oxidative damage (related to DNA, proteins, and lipid) in the airways and a reduction of endogenous antioxidant enzymes [4,5], thus aggravating the oxidant/antioxidant unbalance [6]. Indeed, oxidative stress, and the consequent oxidative damage, through the accumulation of lipid peroxidation products (Malondialdehyde (MDA)) and DNA oxidation (8-hydroxy-2′-deoxyguanosine (8-OHdG)), is responsible for post-translational modification of proteins (carbonylation, nitration), altering mitochondrial regulation pathways in neurodegenerative and cardiovascular diseases, and chronic inflammation in COPD [7,8]. In addition, the higher oxidative burden, associated with inflammatory processes, is accompanied by a continuous cycle of DNA damage and repair, leading to a higher rate of cell turnover, increasing the likelihood of genetic errors [9].

Inflammation, therefore, plays a pivotal role in the pathogenesis of COPD. The presence of chronic pulmonary disease leads to a degeneration of motor activity, caused by alterations in the function of skeletal and ventilator muscles [10,11], causing, in turn, an increase of ROS, leading to muscle damage and inflammation, thus generating a vicious circle of disabling symptoms [12]. Protease-antiprotease imbalance, and the higher number of pro-inflammatory cells (mainly neutrophils), are the main drivers of primarily non-infectious inflammation in COPD. The complex series of events that involves immune trigger cells, and activates innate and adaptive immune responses, contribute to the irreversible airflow limitation, lung remodeling, and emphysema in these patients. COPD clinical features include persistent respiratory symptoms and only partially reversible airflow obstruction, due to an abnormal inflammatory response of the lungs to noxious particles and gases. Exacerbations of COPD, for example, are acute inflammatory events [13,14].

Pulmonary rehabilitation (PR) is considered the key component of the management of COPD patients [15]. A comprehensive intervention, which includes exercise training, education, and behavioral changes, is the most suitable approach to improve muscles endurance and strength and reduce symptoms of dyspnea [16]. A regular exercise program could be considered a good oxidative defense system, improving adaptive responses, promoting a more efficient oxidative metabolism, and/or increasing the activation of endogenous antioxidant systems [17]. ROS excess is inactivated by antioxidants (such as polyphenols) which inhibit molecule oxidation by removing free radicals [18]. Recently, the use of natural antioxidants in the treatment of COPD because of their protective role against oxidative stress has been described [5,19]. The intake of nutrients and micronutrients with antioxidant properties plays an important role in DNA metabolism and DNA repair, contributing to maintain genome stability, and counteracting the effects of the oxidative stress. Lower levels of DNA damage have been associated with lower levels of several inflammatory diseases severity and moderate exercise training (Figure 1) [9,20,21].

High consumption of fruit and vegetables has been associated with a lower risk of lung diseases [22,23]. It is considered a key intervention for the prevention and treatment of cardiovascular diseases [24], and it is associated with a lower risk of cancer and all-cause mortality [25]. Vegetables, and especially green leafy and beetroot, are the primary dietary nitrate source in our diet. Studies on acute nitrate supplementation have shown improved exercise performance, increasing walking distance [26,27,28,29], and lower blood pressure in COPD patients. Moreover, a daily vegetables rich diet showed better FEV1 (Forced Expiratory Volume in the 1st second), compared to a free choice diet [30]. Epidemiological studies highlighted a positive correlation between fruit and vegetable intake, COPD, and lung function [4,5,31,32,33]. In particular, a high intake of fruit and vegetables may protect the lungs from oxidative damage, through their antioxidant and anti-inflammatory properties, due to the high quantity of antioxidants, vitamins (particularly vitamin C, E), β-carotene, minerals, and fiber. In observational studies conducted in teenagers and in middle-aged adults, circulating levels of inflammatory mediators, such as tumor necrosis factor α (TNFα), interleukin (IL-) 6, and IL-17A were inversely correlated with fruits and vegetable intake. A recent study reported that high vegetable intake, particularly green leafy and cruciferous vegetables, correlated with lower levels of white blood cell counts [34]. The anti-inflammatory effect of vegetable assumption could be explained considering their content in polyphenol compounds with antioxidant effect, folate, and flavonoids [35], and through their restoration of water loss and electrolyte balance function [36]. Moreover, the high dietary fiber content supports gut health and the growth of microbial species which potentially modulate the production of pro-inflammatory chemokines and cytokines [37].

In a recent explorative study, we observed an association between daily diet and physical performance in a small group of 36 COPD patients undergoing pulmonary rehabilitation. Using the distance walked in six minutes (in meters) as the main outcome, we found a possible relationship between fruit and vegetable intake and the distance walked at the beginning of the rehabilitation program (135.71 ± 74.6 for patients with low consumption versus 192.86 ± 39.9 in people with high consumption) (unpublished results). Prompted by these data, we decided to further investigate the relationship between fruit and vegetable consumption and the rehabilitation response to verify whether there was a link. Thus, the objective of this study was to evaluate, based on high or low consumption of vegetables, the response of patients to pulmonary rehabilitation. Clinical parameters were integrated with cellular and molecular measurements to obtain a better and personalized treatment option for COPD patients.

## 2. Materials and Methods

### 2.1. Study Design and Participants

An observational cohort study was carried out in 105 patients aged 70 years or older suffering from severe COPD and admitted to the Pulmonary Rehabilitation (PR) Unit of the IRCCS San Raffaele Roma between January 2013 and December 2015 for a comprehensive 3 weeks PR program. Peripheral blood samples were collected and stored at −80 °C at admission and after 3 weeks of PR. Additional detail of the study population can be found in Russo et al., (2019) [19]. The study was approved by the ethics committee of the IRCCS San Raffaele Roma (Prot. 15/2013), and all participants signed the consent to participate in the study at admission.

All patients at admission received a European Union (EU) validated questionnaire to estimate food items intake [38]. Given the robust evidence demonstrating beneficial effects starting from a daily consumption of vegetables [39], we compared patients eating vegetables once a day or more frequently (higher intake) with those reporting a low/moderate intake of vegetables (from 4 times a week up to a minimum of less than once a week). Nobody answered ‘never’ and 25 patients did not answer.

### 2.2. Rehabilitation

All patients received daily inhalation treatment with corticosteroids (beclomethasone dipropionate [C_28_H_37_ClO_7_] (0.4 mg/mL), in combination with bronchodilators, i.e., SABA (Short Acting Beta2 Agonists, in our patients Salbutamol) and SAMA (Short-Acting Muscarinic Antagonist, in our patients anti sub type M3: Ipratropium bromide) for 3 weeks. A questionnaire was administered at admission with questions about demographics informations, medical history, life-style, including a frequency of intake of selected food items. Anthropometric measures, cognitive function, and the long-term use of oxygen therapy were recorded as well. An instrumental evaluation of spirometry, pulse oximetry, blood pressure, electrocardiography (ECG), cardiac frequency, the disease-specific respiratory status (Medical Research Council, MRC), the Barthel and the Borg scales for assessing dyspnea, and the functional exercise capacity (six Minutes Walking Test (6MWT)), was assessed at baseline and at discharge. The choice of the 6MWT distance threshold, indicating the response to treatment is generally considered the most reliable clinical outcome. The distance of ≥30 m, especially in advanced COPD patients, has been indicated by many as a clinically significant response [40]. Forty-four patients scored zero at admission, and 32 of them scored zero also at discharge.

### 2.3. Alkaline Comet Assay

The complete detailed procedure for the assay can be found in our previous work [19]. In summary, following new guidelines for minimal required detailed information on reporting comet assay procedure [41] and new technical recommendations [42,43], damageof DNA from lymphocytes was evaluated after lymphocytes lysation, DNA denaturation, electrophoresis on agarose gel and staining, using the Comet assay IV software (Instem, London, UK). Tail intensity values (TI, % DNA in comet tail) were calculated from 100 comets counted for each individual.

### 2.4. Markers of Oxidative Stress

The following markers of oxidative stress were measured to evaluate the difference associated with vegetable consumption or response to treatment.

#### 2.4.1. Malonaldehyde (MDA) Assay

MDA quantification was performed through thiobarbituric acid reactive substances (TBARS) assay in plasma. Briefly, plasma samples were transferred to a vial containing 10% NaOH, 20% Acetic Acid and TBA. Vials were boiled at 95 °C for 1 h, and then were placed on ice to stop the reaction. Samples were centrifugated 10 min at 1600× *g* at 4 °C and then were placed in a black 96-well microtiter plate. MDA-TBA adduct was measured spectrophotometrically at 530 nm using Tecan sunrise (Tecan).

#### 2.4.2. 8-Hydroxy-2′-deoxyguanosine (8OHdG) Assay

8-hydroxy-2′-deoxyguanosine (8OHdG) was performed using an immune-competitive assay (8-hydroxydeoxyguanosine (8-OHdG), Biomatik ELISA kit, Biomatik, Wilmington, DE, USA) in plasma, following the manufacturer’s protocol. The absorbance was measured at 450 nm using Tecan sunrise (Tecan). The values obtained were expressed as pg/mL.

### 2.5. Markers of Inflammation

Interleukin 6 (IL-6) levels were determined by Enzyme-Linked Immunosorbent Assay (ELISA) in plasma, following the manufacturer’s protocols, using the Human IL-6 DuoSet ELISA (R&D Systems, Minneapolis, MN). C Reactive protein was determined by latex-enhanced immunoturbitimetry using a reference value of less than 0.5 mg/dL. C reactive protein was not included in the statistical analysis since it was measured only in a small group of patients (28 at admission and 12 at discharge) affected by acute respiratory symptoms.

### 2.6. Blood Test

The number and percentage of peripheral blood cells were evaluated during routine laboratory analysis using standard blood count automated methods (Beckman Coulter LH500). LMR, NLR PLR were calculated for each patient diving the absolute cell numbers of the indicated subpopulation, taken from complete blood count. Subsequently, the mean and standard deviation for observed values has been calculated. An increase in NLR and in PLR, as well as a decrease in LMR are indicative of an ongoing inflammation [44,45,46].

### 2.7. Statistical Methods

Descriptive statistics of epidemiological, clinical, and biological data were represented as the frequency with percentage, mean with standard deviation for categorical and continuous variables, respectively. Group differences in demographic, clinical, and laboratory datawere assessed using the Student’s *t*-test or the Mann-Whitney *U* test depending upon the distribution of variables. For the comparison of qualitative variables, the χ^2^ test was used. The adjusted comparison between responders and non-responders and the test for detecting interaction were carried out through the unconditional logistic regression model. The Likelihood Ratio Test was applied to assess the significance of each variable in the logistic model. To allow the calculation of Odds Ratios in the 2 × 2 tables with one zero cell the Haldane-Anscombe correction was applied [47]. A statistically significant value was considered *p* ≤ 0.05. All statistical analyses were done using the statistical software STATA and SPSS (version 26.0).

## 3. Results

A total of 105 elderly patients (mean age ± SD: 72.8 ± 8.8 years) were recruited with severe COPD at hospital admission after signing the informed consent. According to the questionnaire 46 patients were considered at high intake, and 34 at low/moderate intake of vegetables. Patients were moderately overweight (mean BMI: 27.3), and were classified by the response to the treatment as non-responder or responder depending on the increase in their 6MWT distance after three weeks of rehabilitation. Demographic characteristics, lifestyle, and other clinical parameters are reported in Table 1. There were no statisticallysignificant differences in weight, height, BMI, years of education, education level, and marital status among responders and non-responders. No difference was also observed for smoking habits and fruit intake, while the vegetable intake was significantly higher in the group of responders (*p* < 0.05), and responders were significantly younger (*p* < 0.001). COPD patients did not respond differently to treatment depending on the number of comorbidities and oxygen therapy. Responders had a higher Barthel index at admission (*p* < 0.001) and a better improvement at the end of rehabilitation (*p* < 0.05) and started rehabilitation earlier than non-responders (6.0 vs. 10.5 days; *p* < 0.001).

Table 2 shows changes in hematological parameters before and after pulmonary rehabilitation and the relative change (∆) during this period. Patients with a better response to the rehabilitation have slightly more erythrocytes, and a higher hemoglobin concentration, especially at discharge (12.7 vs. 11.3 g/dL, *p* < 0.01). At the end of the rehabilitation program, also neutrophils and lymphocytes have different counts between the two groups. In particular, at discharge, responders had lower neutrophils counts (*p* < 0.05), and higher lymphocytes counts (*p* < 0.02) when compared to non-responders. Responders also showed a significant difference in the variation of eosinophils during treatment (−0.3 vs. 0.5, *p* < 0.01) No difference had been found comparing responders and no-responders for the following parameters: Glycemia, Azotemia, Total Bilirubin, Na^+2^, K^+2^, ALT_GTP, AST_GTP, γGT, pH, pO_2_, pCO_2_, SpO_2_, Heart Rate, Blood Pressure Systolic, Blood Pressure Diastolic (data not shown).

When analyzing biomarkers of oxidative stress, inflammation, and DNA damage we observed that COPD patients, who positively responded to treatment, showed a significantly lower erythrocyte sedimentation rate (ESR) at admission (*p* < 0.05), a difference that remained at the end of the three weeks of rehabilitation (*p* < 0.05). Responders who started rehabilitation with a lower level of inflammation (ESR = 28.6) further reduced this parameter at discharge (ESR = 23.9), while non-responders in the same period increased ESR level, from 43.1 to 49.3 (Table 3). The apparent contrast with the ∆ score reported in the table may be attributed to the smaller number of subjects with data on both admission and discharge. A similar pattern was shown by other biomarkers of oxidative stress and inflammation, with consistently higher values of IL-6, 8-OHdG, and MDA in the group of non-responders when compared to the group of responders (although with *p* = 0.054). A more evident difference between the two study groups was found in cellular markers of inflammation, with the group of responders showing at discharge a significantly higher lymphocytes/monocytes ratio (LMR, *p* < 0.05), a lower neutrophils/lymphocytes ratio (NLR, *p* < 0.05), and a lower platelets/lymphocytes ratio (PLR *p* = 0.01),compared with non responders. High MLR and low NLR and PLR suggest a lower level of inflammation. Results of the comet assay with the two groups showed comparable levels of DNA damage.

In Table 4 is reported a parallel analysis comparing the level of oxidative stress, inflammation, and DNA damage in COPD patients divided by vegetable intake. A comprehensive evaluation of all biomarkers showed that subjects with lower daily consumption of vegetables had a higher level of oxidation, although in some cases statistical significance could not be achieved. The ratio of Lymphocytes/Monocytes (*p* < 0.01), Neutrophils/Lymphocytes (*p* < 0.02), and Platelets/Lymphocytes (though supported by a borderline statistic significativity (*p* = 0.07)) showed consistently higher levels of inflammation in patients with a lower intake of vegetables. Other markers of inflammation showed similar results, although only IL-6 levels at the end of rehabilitation reached a nearly significant performance (*p* = 0.05). The amount of oxidative stress, measured with the malondialdehyde assay, was higher in patients with a lower vegetable intake (*p* < 0.066), while the genomic instability measured with 8-OHdG and with the alkaline comet assay did not show significant differences, although a borderline significant increase of DNA damage was observed during treatment in patients with a lower intake of vegetables.

The multiple logistic regression model built to remove the hypothesis that the association between vegetable intake and response to rehabilitation could be explained by the observed difference by age and Barthel index at baseline showed an Odds Ratio of 3.6 (95% CI 0.96–14.1) to be a responder for those patients reporting a high vegetable intake, although this result remained only borderline significant (*p* < 0.057). When the presence on interaction between age-class and vegetables intake was tested, a significant interaction term was found for patients in the age-class ≥80 years (β = 2.615, SE = 1.117; *p* = 0.0199), clearly showing as the positive effect of vegetables was concentrated in the oldest old, a group which is well known to have a poor consumption of vegetables. These findings are reported in Table 5, which describes the association between diet and rehabilitation performance stratified by age-class.

## 4. Discussion

The results of this study showed a marked improvement in most clinical and functional parameters of COPD patients after three weeks of PR. Our results, in line with literature, showed that high vegetable intake regularly assumed with normal diet, without any specific intervention, could improve the response to rehabilitation therapy, even in patients affected by severe COPD.

Considerable evidence in literature support a crucial role of oxidative stress and associated inflammation in COPD patients; indeed, if compared to healthy subjects, COPD patients had a significant increase in markers of oxidative stress and ROS-induced DNA damage [4,5]. Specifically, oxidative stress occurs when free radicals (reactive oxygen and nitrogen species) exceed the availability of antioxidants, thus causing pathological reactions responsible for numerous diseases, including respiratory ones, and in particular, COPD. Elevated ROS production is directly responsible for increased in lipid peroxidation products (MDA), DNA damage, and the onset of inflammation. The interplay between oxidative stress, oxidative DNA damage, and inflammation, how these events are associated with the success of rehabilitation, and the role of dietary intake of antioxidants is summarized in Figure 1 and is the aim of this study.

The prolonged inflammatory status, induced by the chronic low-grade inflammation, plays a critical role in COPD, inducing irreparable damage to tissues and organs, increasing the risk of disease, and favoring its progression. IL-6, a product of the ongoing inflammation in the airways, is a key modulator of the overall immune response and non-immune cells function [48]. This cytokine can modify the cellular apoptotic response since when cells are exposed to oxidizing agents, elevated levels of IL-6 block proapoptotic pathways throughblc2 or mlc-1) and also promote cell proliferation [49]. Lymphocytes/monocytes ratio (LMR), neutrophils/lymphocytes ratio (NLR), and platelet/lymphocytes ratio (PLR) are rather novel inflammatory markers. They have gained enormous reliability in many chronic inflammatory diseases, starting from the first description of NLR increase in cancer patients in 2001 [50] and, interestingly, represent a good marker of exercise-induced inflammation [51]. Taken all data together, seems that oxidative stress can initiate the inflammatory process, and through the modification of lipids, proteins, and nucleic acids, it can also maintain the disease state by creating a vicious circle (Figure 2).

Evidence emphasizes the importance of diet as a modifiable risk factor for COPD and that some adverse functional consequences of severe COPD are reversible by nutritional support [5]. A recent study suggested that a dietary nitrate supplementation can improve the effect of pulmonary rehabilitation [29]. Moreover, according to different systemic reviews, meta-analysis of prospective cohort and cross-sectional studies, and other analysis and reviews of the dietary intervention studies on COPD patients [52,53], it has been suggested that there is inverse and independent association between high long-term consumption of fruits (in both men and women) and vegetables (only in men) and 35% lower risk of COPD incidence in men, and 37% lower risk in women [52,53]. Consistent evidence has also been reported also for DNA damage and genomic instability [9].

The results of the present study, confirmed to a great extent the model featured in Figure 1, even if in some cases without the support of statistical significance, due to the relatively small number of patients. For the sake of simplicity, patients admitted to rehabilitation were classified as responders or non-responders according to the presence of a clinically significant improvement in the number of meters walked in 6 min, a generally accepted and comprehensive index of rehabilitation performance. The presence of an improved effect of rehabilitation in patients reporting a high intake of vegetables, observed in a pilot study on COPD patients (unpublished results), was confirmed by the results reported in Table 1. The evaluation of several steps of the pathway linking diet to rehabilitation performance is a major strength of this study.

The intake of antioxidants, through a diet rich in vegetables, had a beneficial effect on the mechanisms described in Figure 2, as antioxidants influence DNA metabolism and DNA repair, help to maintain the stability of the genome, and counteracts the effects of oxidative stress. Cellular markers of inflammation LMR and NLR showed a strong and significant difference associated to the response to treatment, while ESR, IL-6, MDA, and the comet assay although showing no or little significative difference were consistent in reporting higher indexes of damage in subjects with a lower response to rehabilitation programs.

Surprisingly, the ROS-induced DNA damage, measured with 8OHdG and partially with the alkaline comet assay, in COPD patients, either before or after three weeks of pulmonary rehabilitation, failed to show major differences associated with dietary intake of vegetable or response to rehabilitation (finding already reported by Russo et al., 2019 [19]). Furthermore, Mercken et al. in 2005 [12], also showed that COPD patients at eight weeks of rehabilitation had no significant increases in ROS-induced DNA damage, unlike what occurred immediately after peak exercise. These data are comparable with the variation in oxidative stress, measured by MDA assay in the plasma of COPD responder and non-responder patients during rehabilitation. Indeed, even in this case, we did not observe any significant variation in MDA levels between responders and non-responders, either before or after rehabilitation treatment. A possible explanation of these findings is that patients who received three weeks of pulmonary rehabilitation probably have not yet reached a sufficient training state to counteract the values of DNA damage and oxidative stress induced by ROS.

From a comprehensive reading of these data, there is evidence that COPD patients with high vegetable intake had a slight reduction in MDA levels and DNA damage, and a more remarkable decrease in inflammatory status, as confirmed by the decrease of blood parameters NLR and LMR. This observation is also crucial for the prevention of COPD exacerbations, given that it has been shown that high NLR (>3.54) is a strong predictor of the risk of developing exacerbations [54]. Therefore, it can be speculated that a high vegetable intake could also help in the prevention of exacerbation in COPD patients.

In recent years, the control of inflammation and DNA damage-associated oxidative stress, together with nutrients intake (as an important part of non-pharmacological strategies, usually with/without physical exercise included), is gaining more and more attention for their possible role in strategies to prevent exacerbations and contrast disease progression. Currently, the standard of care is mostly based on pharmacological interventions, including short-acting bronchodilator drugs (prescribed to patients for immediate relief) and long-acting muscarinic receptor antagonists, beta-2 agonists, or corticosteroids [55]. The difference in the vegetable intake between responders and non-responders, together with the general improvement in inflammatory levels among COPD patients with high consumption of vegetables and after three weeks of rehabilitation treatment, raises the point of a future consideration of these parameters in the prevention and rehabilitation of COPD patients. 

The presence of interaction between vegetable intake and age-class in moderating response to rehabilitation revealed as a high vegetables intake in older patients is more effective than in younger, despite the lower consumption of vegetables in oldest age-classes, clearly shown by Table 5, but also well-known in the literature [56]. A systematic review showed a positive association between fruit and vegetables intake (separate and together) and several measures of muscle strength and function, i.e., walking speed, chair rise, Timed Up and Go Test (TUG), Senior Fitness Test, grip strength, and leg extension strength in a population aged over 70 years. These findings have been challenged by the scarce evidence from intervention studies concerning the effect of a diet rich in vegetables on muscle strength and sarcopenia [57]. The mechanisms through which a diet rich in vegetables may offer myoprotection are not completely understood, it is possible that different constituents in vegetables may have anti-oxidative and anti-inflammatory properties, potentially enhancing muscle function. Thus, it is possible that different nutrients and non-nutrients present in vegetables such as fibers, vitamins, minerals, plant sterols, polyphenols, flavonoids, and alkaline salts, may increase the total exogenous anti-oxidative ability, counteracting the pro-inflammatory response in ageing muscle. Moreover, the phytochemicals content of vegetable such as flavonoids and non-flavonoids, as well as sugars, acids and polysaccharides are known to improve cardiovascular and cardiometabolic health through several proposed mechanisms, e.g., antithrombotic, anti-atherosclerotic, lipid profile and blood pressure regulation, and glucose metabolism [32,58]. A high daily intake of fruits and vegetables in elderly, in the absence of disease, has been associated with higher lipophilic antioxidant levels as well as with the presence of lower levels of biomarkers of lipid peroxidation [59]. The excessive fragmentation of our data, when stratified by response and specific biomarker did not allow a proper evaluation of this possible pathway in a specific age-class.

The present study has several strengths. The most remarkable is the observed consistency between biomarkers findings and the candidate mechanism, based on the results of the preliminary study and on published evidence. Among the weaknesses, small numbers are the most restrictive. The reference group in our study reported a moderate (weekly) intake of vegetables (only one subject in the whole study group reported eating vegetable less than once per week). The use of a control group with a low to moderate consumption of vegetables, while has reproduced a real-life condition, has probably underestimated the anti-inflammatory effect of this diet.

In conclusion, this study demonstrated that high vegetable intake, without any specific intervention, can identify COPD patients with a higher probability to respond successfully to rehabilitation. In addition, while 3 weeks of pulmonary rehabilitation are probably too short to reveal an improvement in oxidative stress and a reduction of DNA damage, they are long enough to show improvement in the patient’s inflammatory state.

## Figures and Tables

**Figure 1 nutrients-13-02787-f001:**
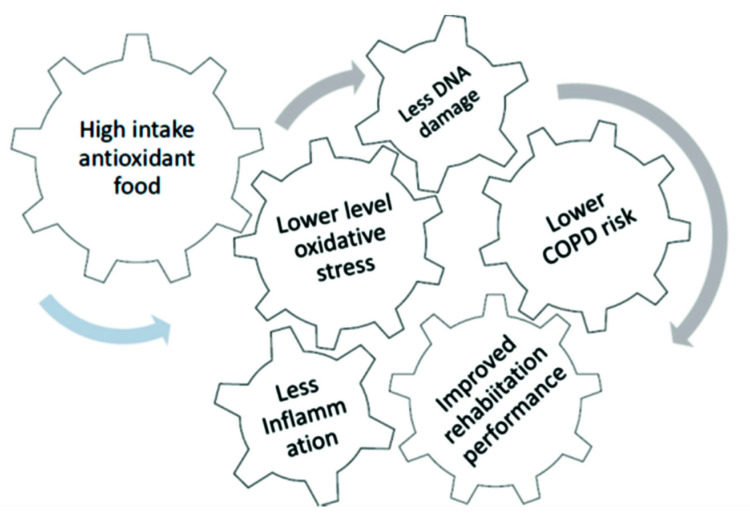
Complex machinery illustrating how a high consumption of food with antioxidant properties may be beneficial in reducing the level of oxidative stress and inflammation, but also improving genomic stability, reducing the risk of COPD, and improving physical performance.

**Figure 2 nutrients-13-02787-f002:**
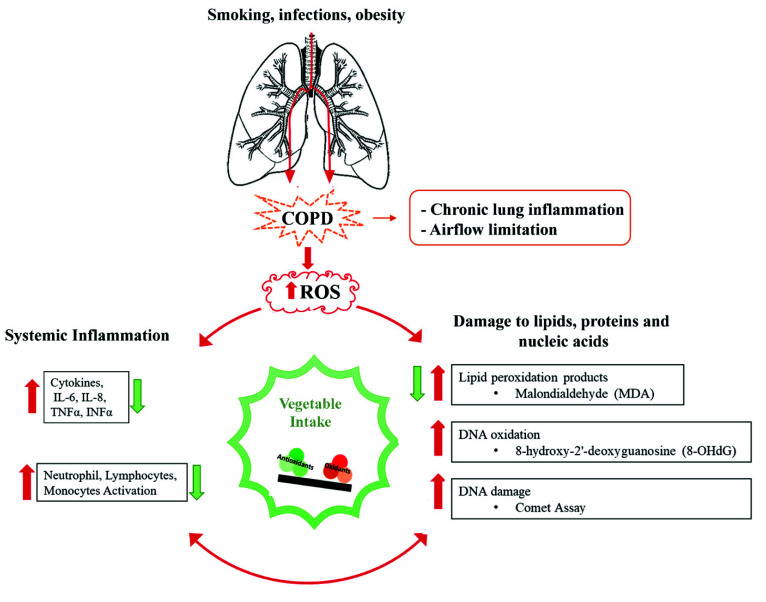
Scheme summarizing this study. Continued production of ROS (red arrow), due to increased oxidative stress, systemic inflammation, and DNA damage (red arrows), is responsible for the development and maintenance of severe COPD. This mechanism is attenuated following the pulmonary rehabilitation program and through a high vegetable intake (green arrows).

**Table 1 nutrients-13-02787-t001:** Baseline characteristics of COPD patients by response to the treatment.

Variable	*N*	Total (105)	*N*	Non-Responder (36)	*N*	Responder (69)	*p*-Value
Sex (male), No (%)	47	44.3	17	47.2	30	43.5	NS
Age-Class, No (%)							
*≤66.2 Years*	26	24.8	5	13.9	21	30.4	
*66.3–73.3 Years*	27	25.7	2	5.5	25	36.2	
*73.4–79.9 Years*	27	25.7	18	50.0	9	13.0	<0.001
*≥80 Years*	25	23.8	11	30.6	14	20.3	
Weight (Kg), Mean (SD)	75	72.5 (21.8)	16	73.7 (23.6)	59	72.1 (21.5)	NS
Height (cm), Mean (SD)	73	163.3 (9.0)	15	166.4 (9.2)	58	162.5 (8.9)	NS
BMI, Mean (SD)	72	27.3 (7.8)	14	25.8 (8.3)	58	27.7 (7.7)	NS
Years of Education, Mean (SD)	105	9.0 (4.0)	36	9.3 (4.5)	69	9.0 (3.7)	NS
Years of Education, No (%)							
*8*	66	62.9	21	58.3	45	65.2	
*9–13*	32	30.5	11	30.6	21	30.4	
*>13*	7	6.7	4	11.1	3	4.3	NS
Marital Status, No (%)							
*Unmarried*	6	5.7	4	11.1		2 (2.9)	
*Married*	55	52.4	23	63.9		32 (46.4)	
*Widow*	28	26.7	8	22.2		20 (29.0)	NS
*Divorced*	16	15.4	3	8.3		13 (18.8)	
Smoking Habit, No (%)							
*Never*	12	12.8	4	13.8	8	12.3	
*Former*	66	70.2	21	72.4	45	69.2	NS
*Current*	16	17.0	4	13.8	12	18.5	
Fruit Intake, No (%)							
*High Intake (daily)*	67	77.0	19	70.4	48	80.0	NS
*Low/Moderate Intake (Weekly)*	20	23.0	8	29.6	12	20.0	
Vegetable intake, No (%)							
*High Intake (Daily)*	46	57.5	10	40.0	36	65.4	
*Low/Moderate Intake (Weekly)*	34	42.5	15	60.0	19	34.6	0.033
Δ from Acute Event (days), Median (IQR)	105	8.0 (9)	36	10.5 (12)	69	6.0 (7)	0.001 ^+^
6MWT, Mean (SD)							
*Admission*	105	86.7 (85.0)	36	---- ^++^	69	124.2 (76.7)	---
*Discharge*	105	175.1 (135.9)	36	----	69	261.6 (72.7)	---
*Δ Score*	105	90.7 (81.6)	36	---	69	137.2 (61.0)	---
Barthel Index, Median (IQR)							
*Admission*	105	76.0 (40)	36	42.0 (33)	69	83.0 (13)	<0.001 ^+^
*Discharge*	103	91.0 (20)	34	66.0 (26)	69	95.0 (9)	<0.001 ^+^
*Δ Score*	103	14.0 (9)	34	23.5 (21)	69	14.0 (11)	0.033 ^+^
Oxigen Therapy, No (%)	33	31.4	13	36.1	20	29.0	NS
Number of Comorbidities, Mean (SD)	105	3.42 (1.71)	36	3.86(1.99)	69	3.13 (1.45)	NS
FEV1/FVC, No, Mean (SD)							
*Admission*	52	47.3 (24.3)	8	31.6 (20.8)	44	50.1 (24.0)	0.085
*Discharge*	32	61.7 (22.7)	4	58.1 (17.0)	28	62.2 (23.6)	NS
*Δ Score*	25	10.1 (9.3)	2	6.2 (5.7)	23	10.5 (9.6)	NS

^+^ Mann-Whitney *U*-test; ^++^ Mean values were not reported because of thelarge number of zeros (32/36).

**Table 2 nutrients-13-02787-t002:** Characteristics of blood test parameters in COPD patients by the response to thetreatment.

Variable	*N*	Total (*n*.105)	*N*	Non-Responder (*n*.36)	*N*	Responder (*n*.69)	*p*-Value
Red Blood Cells, *Mean (SD) 10^6^/μL*							
*Admission*	103	4.4 (0.7)	35	4.2 (0.8)	68	4.5 (0.6)	0.068
*Discharge*	100	4.3 (0.8)	35	4.1 (0.6)	65	4.4 (0.8)	0.074
*Δ Score*	100	−0.1 (0.7)	35	−0.1 (0.5)	65	−0.1 (0.7)	NS
Platelets, *Mean (SD) 10^3^/μL*							
*Admission*	102	245.3 (89.1)	34	242.2 (90.1)	68	246.8 (89.2)	NS
*Discharge*	98	205.7 (78.3)	34	219.5 (89.3)	64	198.4 (71.4)	NS
*Δ Score*	98	−39.3 (78.3)	34	−22.7 (88.6)	64	−48.1 (71.3)	NS
White Blood Cells, *Mean Number (SD) 10^3^/μL*							
*Admission*	103	10.1 (3.9)	35	10.0 (4.6)	68	10.2 (3.6)	NS
*Discharge*	99	9.8 (3.3)	35	10.0 (3.9)	64	9.7 (3.0)	NS
*Δ Score*	99	−0.4 (3.6)	35	−0.03 (4.3)	64	−0.5 (3.2)	NS
Neutrophils, *Mean Number (SD) 10^3^/μL*							
*Admission*	103	7.5 (72.9)	35	7.7 (74.2)	68	7.4 (72.3)	NS
*Discharge*	99	6.9 (68.5)	35	7.6 (72.6)	64	6.5 (66.3)	0.025 ^+^
*Δ Score*	99	−0.7 (4.7)	35	−0.1 (1.5)	64	−0.1 (6.4)	0.056
Lymphocytes, *Mean Number (SD) 10^3^/μL*							
*Admission*	102	1.7 (17.8)	35	1.5 (16.7)	67	1.8 (18.3)	NS
*Discharge*	99	2.0 (21.9)	35	1.6 (18.5)	64	2.2 (23.7)	0.016 ^+^
*Δ Score*	99	0.3 (4.5)	35	0.1 (1.8)	64	0.4 (6.0)	0.051
Monocytes, *Mean Number (SD) 10^3^/μL*							
*Admission*	101	0.7 (7.7)	34	0.8 (8.1)	67	0.7 (7.5)	NS
*Discharge*	99	0.6 (7.6)	35	0.6 (7.6)	64	0.6 (7.6)	NS
*∆ Score*	99	−0.09 (0.03)	34	−0.1 (0.3)	64	−0.08 (0.2)	NS
Eosinophils, *Mean Number (SD)10^3^/μL*							
*Admission*	101	0.1 (1.2)	34	0.10 (1.6)	67	0.09 (1.1)	NS
*Discharge*	99	0.1 (1.5)	35	0.07 (1.1)	64	0.14 (1.6)	0.087
*Δ Score*	97	0.02 (0.2)	34	−0.03 (0.5)	63	0.05 (0.6)	0.006
Basophils, *Mean Number (SD) 10^3^/μL*							
*Admission*	101	0.02 (0.3)	34	0.02 (0.3)	67	0.03 (0.3)	NS
*Discharge*	99	0.03 (0.3)	35	0.03 (0.4)	64	0.03 (0.3)	NS
*Δ Score*	97	0.004 (0.04)	34	0.004 (0.04)	63	0.003 (0.04)	NS
Hemoglobin, Mean (SD) g/dL							
*Admission*	66	12.6 (1.8)	21	12.0 (1.9)	45	12.9 (1.8)	NS
*Discharge*	55	12.3 (1.6)	16	11.3 (1.6)	39	12.7 (1.5)	0.004
*Δ Score*	51	−0.5 (2.1)	15	−1.3 (3.5)	36	−0.2 (1.1)	0.093

^+^ Mann-Whitney *U*-test.

**Table 3 nutrients-13-02787-t003:** Biomarkers of oxidative stress, inflammation and DNA damage in COPD patients by response to the treatment.

Variable	*N*	Total (*n*.105)	*N*	Non-Responder (*n*.36)	*N*	Responder (*n*.69)	*p*-Value
ESR, Mean (SD) mm/h							
*Admission*	97	33.2 (28.0)	31	43.1 (32.7)	66	28.6 (24.4)	0.024 ^+^
*Discharge*	68	31.0 (30.6)	19	49.3 (39.2)	49	23.9 (23.3)	0.013 ^+^
*Δ Score*	65	−5.5 (27.2)	18	−3.9 (35.8)	47	−6.1 (23.5)	NS
IL-6, Mean (SD) pg/mL							
*Admission*	73	66.6 (107.9)	20	90.3 (116.5)	53	57.6 (104.2)	0.054
*Discharge*	69	105.7 (148.4)	20	137.4 (154.8)	49	92.8 (145.4)	0.067
*Δ Score*	59	42.6 (181.3)	15	43.7 (196.6)	44	42.3 (178.1)	NS
8-OHdG, Mean (SD) pg/mL	38	24.7 (8.1)	13	25.6 (8.9)	25	24.2 (7.9)	NS
MDA, Mean (SD)μM	38	47.0 (12.8)	10	51.2 (13.0)	28	45.5 (12.7)	NS
Comet assay (Tail Intensity, Mean (SD))							
*Admission*	87	19.3 (7.2)	27	18.0 (4.4)	60	19.8(8.1)	NS
*Discharge*	86	22.0 (7.3)	26	21.9 (8.2)	60	22.0 (6.9)	NS
*Δ Score*	84	2.4 (6.5)	26	4.0 (5.8)	58	1.8 (6.7)	NS
Lymphocytes/Monocytes							
*Admission*	101	2.6	34	2.2	67	2.9	NS
*Discharge*	99	3.3	35	2.7	64	3.6	0.040 ^+^
*Δ Score*	97	11.5	34	11.0	63	11.8	NS
Neutrophils/Lymphocytes							
*Admission*	102	5.5	34	6.1	67	5.2	NS
*Discharge*	99	4.6	35	6.3	64	3.7	0.013 ^+^
*Δ Score*	98	−1.2	35	3.8	63	−4.1	NS
Platelets/Lymphocytes							
*Admission*	101	185.4	35	207.7	67	174.1	NS
*Discharge*	98	135.6	35	178.2	64	113.1	0.010 ^+^
*Δ Score*	97	−48.1	35	−48.6	63	−47.8	NS

^+^ Mann-Whitney *U*-test.

**Table 4 nutrients-13-02787-t004:** Biomarkers of oxidative stress, inflammation and DNA damage in COPD patients by daily vegetables intake.

Variable	*N*	Low-Moderate Intake (34)	*N*	High Vegetable Intake (46)	*p*-Value
ESR, Mean (SD) mm/h					
*Admission*	33	36.7 (29.2)	40	28.2 (26.7)	NS
*Discharge*	20	33.9 (35.0)	31	28.4 (29.7)	NS
*Δ Score*	20	−4.2 (39.9)	28	−2.8 (21.2)	NS
IL-6, Mean (SD)					
*Admission*	22	70.0 (89.3)	35	66.4 (130.3)	NS
*Discharge*	15	154.7 (151.9)	36	76.4 (127.5)	0.050 ^+^
*Δ Score*	12	78.7 (166.1)	31	22.1 (188.0)	NS
8-OHdG, Mean (SD) pg/mL	16	24.9 (6.8)	15	25.1 (8.0)	NS
MDA, Mean (SD) μM	19	50.4 (13.6)	19	43.6 (11.4)	NS
Comet assay (Tail Intensity, Mean (SD))					
*Admission*	25	20.0 (7.8)	41	19.9 (6.3)	NS
*Discharge*	25	23.5 (5.6)	41	21.6 (7.2)	NS
*Δ Score*	24	3.6 (6.4)	40	1.2 (6.3)	0.086
Lymphocytes/Monocytes					
*Admission*	34	2.4 (1.8)	43	3.0 (2.1)	0.025 ^+^
*Discharge*	32	2.5 (1.1)	43	4.0 (2.3)	0.001 ^+^
*Δ Score*	32	2.6 (11.3)	41	6.1 (30.3)	NS
Neutrophils/Lymphocytes					
*Admission*	34	5.9 (3.8)	44	4.9 (3.0)	NS
*Discharge*	32	5.6 (4.7)	43	3.6 (2.4)	0.017
*Δ Score*	32	−0.7 (21.3)	42	3.6 (18.6)	NS
Platelets/Lymphocytes					
*Admission*	33	186.4 (119.3)	44	166.8 (102.5)	NS
*Discharge*	31	161.5 (143.8)	43	110.8 (99.5)	0.066
*Δ Score*	31	−67.8 (400.9)	42	50.9 (669.2)	NS

^+^ Mann-Whitney *U*-test.

**Table 5 nutrients-13-02787-t005:** Response to the treatment of COPD patients by vegetables intake and age-class.

Class of Age /Vegetables Intake	% of High Vegetables Eaters	Responders/ Non Responders		Odds Ratio	*p*-Value
≤66.2 years					
*High*	78.0%	15	3	1.25	1.000
*Moderate*		4	1		
66.3–73.3 years					
*High*		12	0		
*Moderate*		9	0	1.37	0.645
73.4–79.9 years	59.9%				
*High*		4	7		
*Moderate*		1	6		
≥80 years					
*High*	27.8%	5	0 *	17.0	0.036
*Moderate*		5	8		

* Odds Ratio has been calculated using the Haldane-Anscombe correction [47].

## Data Availability

The data presented in this study are available on request from the corresponding author. The data are not publicly available due to privacy restrictions from the ethics committee.

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
