# Peer review of "Daily Vegetables Intake and Response to COPD Rehabilitation. The Role of Oxidative Stress, Inflammation and DNA Damage"

_nutrients, 2021, doi:10.3390/nu13082787_

Round 1

Reviewer 1 Report

Sara Ilari and Colleagues report that high vegetable intake in normal diet can increase the probability to successfully respond to rehabilitation in patients with COPD. They based their observation on cross-sectional study carried out in 105 patients aged 70 years or older suffering from severe COPD and admitted to the Pulmonary Rehabilitation (PR) Unit. The vegetable intake was estimated based on a questionnaire which included frequency intake of selected food items. The response to the treatment (non-responder or responder) has been defined based on a  distance gained ≥ 30 m after three  weeks of rehabilitation. Authors investigated also markers of inflammation , oxidative stress and DNA damage.

Unfortunately the retrospective and cross-sectional study design, the imbalance of age (non responders were older) and the start of rehabilitation program ( non responders started later than responder) , the “naïve” estimate of vegetable intake (high, low/moderate based on what?) did not support the conclusion of Authors.

Critical points

-Which validated questionnaire Authors used to estimate food items intake?

-Had the questionnaire been administered blinded of results?

-6minWT following rehabilitation has not been reported in the TABLE!

-C Reactive Protein values should be reported

Reviewer 2 Report

The study design is not cross-sectional. It is better described as an observational cohort. 

Differences the Barthel index and age at baseline should be adjusted for in the analyses of change over the 3 weeks.  The higher function at baseline might partially explain the higher response to therapy.

Causal inferences should not be made from these analyses unless specific causal inference methods are applied (and they were not). 

Round 2

Reviewer 1 Report

In the revised manuscript Authors address some of the critical points, i.e. they report the validated questionnaire they used and why they did not report PCR values of their patients. Unfortunately, the retrospective and cross-sectional study design, the imbalance of age (non-
responders were older) and the start of rehabilitation program (non-responders started later than responder), did not support the conclusion of Authors.

Author Response

Thank you 

Reviewer 2 Report

response is adequate.  

Author Response

Thank you